# Cognitive Conflict in Borderline Personality Disorder: A Study Protocol

**DOI:** 10.3390/bs10120180

**Published:** 2020-11-26

**Authors:** Victor Suarez, Guillem Feixas

**Affiliations:** Section of Personality, Evaluation and Psychological Treatment, Department of Clinical Psychology and Psychobiology, Institute of Neurosciences, Campus Mundet, Passeig de la Vall d’Hebron, 171, University of Barcelona, 08035 Barcelona, Spain; gfeixas@ub.edu

**Keywords:** implicative dilemma, dilemmatic construct, cognitive conflict, predictors of outcome, personal construct, repertory grid, borderline personality disorder

## Abstract

Borderline personality disorder (BPD) represents a severe mental condition that is usually characterized by distressing identity disturbances. Although most prevailing explanatory models and psychotherapy approaches consider and intervene on self-concept, they seem not to recognize or explore idiosyncratic cognitive conflicts that patients may experience. These conflicts, which have been conceptualized as “implicative dilemmas” and “dilemmatic constructs” by personal construct theorists, could be considered as key elements of the explanatory model for BPD to provide a better understanding of this disorder and possibly enhance the effectiveness of contemporary psychotherapeutic approaches. The current study (Identifier: NCT04498104) aims to examine the characteristics of the interpersonal cognitive system of a group of patients diagnosed with BPD, using the repertory grid technique, and to compare them with those of a community sample. We will test if BPD participants are more affected by cognitive conflicts than controls. Additionally, we will gauge the association between cognitive conflicts and symptom severity as well as their predictive capacity of treatment outcome. The obtained results will be a necessary step to determine if cognitive conflicts have a substantial role on the explanation of BPD. It could also help to consider the development of a conflict resolution intervention module for this disorder.

## 1. Introduction

Borderline personality disorder (BPD) is a highly disabling condition that has a severe impact on quality of life and psychosocial functioning of those affected [1]. Its prevalence is estimated in 1–2% of the general population, although this rate can go up to 22% in clinical settings [2].

People with BPD may experience a broad variety of clinical manifestations but most of them are related to emotional, interpersonal and self-image instability, along with extreme and impulsive reasoning and demeanor. It is generally understood that mood, sense of identity and relationships tend to drastically fluctuate depending on different factors, causing psychological distress that is intended to be alleviated by responding to the emerging urges with little or no reflection on the consequences [3]. According to the fifth edition of the Diagnostic and Statistical Manual of Mental Disorders (DSM-5) [4], an individual should meet five out of the nine possible criteria to receive a BPD diagnosis. Therefore, it is usually said that two patients with a diagnosis of BPD might have quite different experiences due to the wide diversity of possible symptom combinations they can undergo. On the other hand, the prototypical patient does not always tend to seek assistance to mitigate BPD symptoms, but rather to find relief from comorbid syndromes such as depression, anxiety or eating disorders, among others [5]. This lack of syndromic unity has prompted the creation of dimensional models of personality disorders. As a new feature, this latest version of the DSM includes in its third section a categorical-dimensional hybrid proposal that understands all personality disorders as a single entity composed of five broad domains and 25 specific traits instead of several qualitatively different mental conditions [4].

It is common for individuals with a BPD diagnosis to engage in suicidal and parasuicidal ideation or behavior and substance abuse as a consequence of their main symptoms [3]. It has been generally reported that up to 90% of BPD patients develop a pattern of nonsuicidal self-injuries (NSSIs), such as cutting or burning [6], and about 60–70% commit at least one unsuccessful suicide attempt [7], although other studies find even greater prevalence rates (e.g., [8]). It is estimated that approximately 10% of people with BPD lose their life to suicide [9], increasing by 50 times the suicide probability compared to non-clinical population, which makes BPD one of the major life-threatening mental illnesses [10]. Thus, the need to find effective therapeutic strategies that can diminish these rates deserves critical attention from researchers, clinicians and health authorities.

Studies of up-to-date therapeutic resources for BPD show that pharmacological treatment is only indicated in certain cases, mainly when directly proposed to address certain features of the whole clinical picture [11]. Therefore, psychotherapy is one of the main resources for mental health practitioners when treating these patients [12]. Currently, there are several psychological treatments and explanatory models designed or adapted for people diagnosed with BPD but not all of them have been equally supported by empirical evidence. Although none of the existing proposals has proven to be clearly superior to the others, dialectical behavior therapy (DBT) [13,14,15] and mentalization-based treatment (MBT) [16,17] seem to stand out in their therapeutic effectiveness thanks to the favorable data reported in multiple studies [18]. These approaches come from different psychological models that focus on different aspects when explaining BPD, the former from a cognitive-behavioral base and the latter from a psychodynamic approach. Nonetheless, they also integrate elements and resources from other different psychological orientations and currents of thought. Although they have been reported to be useful in many cases, after detailed analysis, these therapies tend to be successful mainly for certain symptomatic domains [18], leaving a remarkable need to focus on other unattended clinical necessities of people with BPD [19]. Perhaps the explanatory models from which these psychotherapies emerge might have underestimated important aspects, such as the level of conflict in the cognitive system of these patients.

To advance in the understanding of BPD and further improve these psychotherapies or any other therapeutic strategies, perhaps it could be useful to complement them with contributions from a constructivist base. Cognitive conflict [20] is one of the concepts that could be enriching from this perspective. The essence behind this term is the simultaneous presence of two or more cognitive contents (thoughts, beliefs, values, etc.) that are opposed or contradictory and therefore may create internal tension that result in various forms of psychological distress. It is based on personal construct theory (PCT) [21] and it might explain the lack of psychological stability, which is characteristic of people with BPD. Considering that personal identity, interpersonal relationships and affectivity are very unstable domains for these people, we could expect to find high levels of inner conflict that might hinder balance in these areas due to the presence of personal dilemmas related to the construction of the self and others. If that is the case, this issue should be addressed by mental health professionals. Though some psychotherapeutic approaches may deal with some sorts of cognitive conflicts, their content and structure are usually formulated with constructs derived from the theories on which they are based rather than coming from the personal terms and experience of each patient. In the treatment of BPD, it is not usual to perform a systematic assessment of cognitive conflicts and their role in the actual clinical manifestation of the disorder. However, dilemma-focused therapy, a constructivist therapy based psychological intervention, undertakes a systematic exploration of the patient’s cognitive system to search for idiosyncratic dilemmas. Results of this careful assessment are used to formulate a precise clinical hypothesis and treatment plans for each patient. As BPD is a heterogeneous condition with multiple possible diagnostic criteria combinations, adapting to each patient and exploring their construction system and personal cognitive conflicts would be more beneficial than to attribute general, potentially erratic or general assumptions derived from a given academic or theoretical bias [22]. Idiosyncratic inner conflict is a very unexplored field that could potentially be important in the treatment of BPD.

The repertory grid technique (RGT) is a complex assessment instrument created in the context of PCT and used to evaluate the individual’s personal constructs (bipolar dimensions of subjective meaning) and their belief system in order to have access to their sense of identity and interpersonal perception [23,24]. Using this instrument, it is possible to measure cognitive conflicts in the form of implicative dilemmas (not performing a desired change due to fear of changing one’s desired identity aspects) and dilemmatic constructs (not knowing how oneself wants to ideally be), among other cognitive indicators [25]. This tool could be useful to explore the world of personal meanings of patients with BPD in order to facilitate the detection of the cognitive conflicts they may experience as has been done in other clinical populations [26]. Adding a dilemma-focused intervention [27,28] to the existing treatments for BPD could optimize the psychological treatment for this type of patient, as has been already done with depression [29].

## 2. Objectives and Hypothesis

The central aim of this research is to assess cognitive conflicts in individuals with a diagnosis of BPD to determine their role in the explanatory model of this disorder. These findings would permit the consideration of adapting a dilemma-focused intervention module for these patients. In addition, it would be interesting to explore other characteristics of the construction of self and others that are assessed with the RGT. The specific objectives of this study are:To verify the hypothesis that patients with BPD present more cognitive conflicts (i.e., implicative dilemmas and dilemmatic constructs) than a sample from the general population.To explore the content of cognitive conflicts in patients with BPD.To examine if the presence and number of cognitive conflicts are associated with severity of emotional symptoms in BPD patients.To test whether the presence and number of cognitive conflicts has any capacity to predict treatment outcome (in terms of psychological distress and BPD symptomatology).To explore the relevance of other aspects of the construction of self and others to explain the psychological functioning of patients with a BPD diagnosis.

It is expected that:The percentage of participants with implicative dilemmas and/or dilemmatic constructs will be superior in the group of patients diagnosed with BPD compared to a control group (community sample).The number of implicative dilemmas and/or dilemmatic constructs will be higher in the BPD group than in the control group.The presence and higher number of cognitive conflicts will be associated with greater levels of general distress clinical symptomatology (such as depression, anxiety, etc.).The presence and higher number of cognitive conflicts will predict poor treatment outcome one year after the initial assessment.

For exploratory purposes, we will also study the content of implicative dilemmas and dilemmatic constructs, and the differences with the control group regarding self-construction measures (self-ideal discrepancy, self-others discrepancy, ideal-others discrepancy) and other characteristics of the construction system (differentiation and polarization) and its association with other clinical and sociodemographic measures.

## 3. Materials and Methods

### 3.1. Design and Sample

We embrace both quantitative and qualitative methods in this study. For the former, we use a case–control design comparable to the ex post facto prospective simple design with one independent variable (presence of disorder). For the latter, we explore the themes used by participants in their cognitive conflicts using the content analysis of their personal constructs. Using the sample size calculation program GRANMO v.7.12 (Institut Municipal d’Investigació Mèdica, Barcelona, Spain), we found that by accepting an alpha error of 0.05 and a beta error of 0.2 in a two-sided test, 78 subjects are necessary in each group to detect a small effect size between clinical and control groups, assuming a standard deviation of 1. The clinical sample will be formed by participants diagnosed with BPD receiving treatment from outpatient or inpatient public and private mental health facilities in the local area of Barcelona, Spain (CSMA *Germanes Hospitalàries*, ITA-*especialistas en salud mental*, and *ITLímit*). At the time of the first assessment, participants will be undergoing treatment as usual in each respective center, which will at least include individual psychotherapy. Non-specific or specific BPD group therapy, such as DBT or MBT, will also be part of therapeutic program of some participants. Regarding the qualitative line of this project, a content analysis system will be employed for the personal constructs of each participant, with special attention to those constructs that take part in cognitive conflicts.

#### 3.1.1. Inclusion Criteria

Individuals between 18 and 60 years old and with a diagnosis of BPD made by a well-trained professional according to DSM-5 criteria are eligible for the clinical sample. The control group is formed by healthy volunteers recruited in previous studies from 2012 to 2015, who were not receiving psychotherapy at the time of assessment. These participants will be retrieved from a student sample from the University of Barcelona [30] and a nonstudent community sample [31] and will be matched by age, gender and educational level with the clinical sample.

#### 3.1.2. Exclusion Criteria

We exclude from this study potential participants that have been diagnosed with bipolar disorder, psychotic disorder, persistent and active substance abuse, disabling physical illness, organic cerebral dysfunctions or severe mental developmental difficulties. The presence of other comorbid conditions, such as other personality disorders, anxiety disorders, eating disorders, depression or non-disabling physical illness, will not be excluded but will be recorded for statistical control. Finally, it will not be suitable to include those who do not have an enough linguistic competence to communicate in either Spanish or Catalan.

### 3.2. Instruments and Measures

#### 3.2.1. Sociodemographic Questionnaire

A sociodemographic questionnaire is used to collect data for variables such as age, gender, civil state, number of children, profession, level of education and clinical history.

#### 3.2.2. Structured Clinical Interview for DSM-IV-II (SCID-II)

The Structured Clinical Interview for DSM-IV-II (SCID-II) is a structured assessment interview aimed to identify the potential presence or absence of the personality disorders defined by the DSM-IV criteria [32].

#### 3.2.3. Diagnostic Interview for Borderlines—Revised (DIB-R)

The Diagnostic Interview for Borderlines—Revised (DIB-R) is a structured interview aimed to identify the potential presence of BPD by assessing 22 common statements for BPD patients [33,34]. It explores aspects related to affectivity, cognition, impulsivity patterns and interpersonal relationships. Zanarini et al. [35] found high levels of both inter-rater (K = 0.94) and test-retest (K = 0.91) reliability.

#### 3.2.4. Repertory Grid Technique (RGT)

The repertory grid technique is a personalized procedure using a structured interview designed to make explicit personal constructs that interviewees use to interpret and organize their interpersonal world [23,24]. It begins by identifying significant family and friends (here termed elements) and then the patient is asked to compare them (including “self now” and “ideal self”) in pairs in order to elicit the constructs used to differentiate among these elements. Once the interviewee offers a verbal tag to describe an element (e.g., “active”), they are asked to utter the opposed meaning (e.g., “passive”) to complete the bipolar dimension represented by the personal construct. Finally, the patient is asked to rate each relevant person of their interpersonal world (in addition to their actual and ideal selves) according to each personal construct expressed using a seven-point Likert scale. This creates a data matrix of elements with constructs which can be analyzed to extract several cognitive indices.

As previously mentioned, one of the main measures the RGT can provide is cognitive conflict, through the identification of implicative dilemmas and dilemmatic constructs. In order to detect the presence of implicative dilemmas, first we need to identify both congruent and discrepant constructs. The former ones are defined by the congruence between the scores of the “self now” and “ideal self” elements in that construct (e.g., being a “protector” in the example of Feixas et al. [36]), and the latter by a dissonance (or dissimilarity) between these two elements, indicating a discrepancy in the way individuals perceive themselves and how they would desire to be (e.g., not “loving herself” but wishing to do it, in the same case example). An implicative dilemma is detected when a correlation between the scores of a congruent construct and a discrepant construct is found, when the direction that the desired change indicated by the discrepant construct is associated with the loss of congruence for the self (e.g., “loving herself” is associated with being “unemotional”, the opposite pole of being a “protector”). Therefore, discrepant constructs per se do not define a conflict or dilemma but are just a discrepancy (see other measures below), a target for change. 

“Rather, it is the conflictive association between a discrepant and a congruent construct which causes conflict. In these cases, the need for change (she wants to love herself) might be hindered by the need for self-ideal congruency (continue being protective). What an implicative dilemma tells us is that the need for change expressed by the discrepant construct is in conflict with the need for coherence expressed by the congruent construct. Thus, the patient unwittingly hesitates in taking a clear course of action because striving for loving herself has negative implications for her identity. In the view of such a dilemma, change may be less likely to occur because abandoning the symptoms would result in invalidation of core aspects of the self.” [36] (p. 372)

On the other hand, dilemmatic constructs are detected when the patient does not have a preference for neither pole of a personal construct and therefore indicates a middle point in the rating for the ideal self. In this situation, the individual may have doubts regarding the best position they could take in a specific personal construct, thus experiencing another type of dilemma in a construct for which each pole might involve both positive and negative implications for the self. Examples of implicative dilemmas and dilemmatic constructs can be found in Feixas et al. [36] and Paz et al. [37]. 

Other than these types of cognitive conflicts, the analysis of the RGT data matrix provides the following measures:Self-discrepancies: These measures are calculated through Euclidean distances between the elements “self now”, “ideal self” and “others” (considered as the mean of all elements other than the selves).
○Self-ideal discrepancy: Higher distances between “self now” and “ideal self” are usually interpreted as low self-esteem.○Self-others discrepancy: While high scores indicate perceived social isolation, low scores point out a high level of identification with others.○Ideal-others discrepancy: Conceived to measure perceived adequacy of significant others, large distances indicate a critical view of others.Interpersonal dichotomous thinking: Computing the total amount of extreme scores (“1” and “7”) yields the polarization index. This measure reveals if there is a tendency towards dichotomous thinking (all or nothing) when construing self and others.Interpersonal cognitive differentiation: This domain is measured through the percentage of variance accounted for the first factor (PVAFF), an index that results from the correspondence analysis of the whole grid data matrix. A large size of the first factor indicates a tendency to interpret interpersonal experiences in a unidimensional way, while a low PVAFF allows for a more differentiated view, with more nuances, suggesting that multiple points of view are available when construing self and others.

#### 3.2.5. Depression Anxiety Stress Scales (DASS-21)

The Depression Anxiety Stress Scales (DASS-21) is a questionnaire created to quickly evaluate levels of depression, anxiety and stress. Respondents must score on a scale from 0 to 3 to describe the intensity or frequency they experienced in 21 different clinical symptoms during the previous week [38]. These items are grouped in three scales (one for each clinical domain) that reflect the severity of the clinical symptoms with punctuations that oscillate between 0 and 21. The DASS-21 was adapted to the Spanish population by Bados et al. [39] and they reported a Cronbach’s alpha of 0.84 (Depression scale), 0.70 (Anxiety scale) and 0.82 (Stress scale) for each of the domains.

#### 3.2.6. Clinical Outcomes in Routine Evaluation—Outcome Measure (CORE-OM)

Clinical Outcomes in Routine Evaluation—Outcome Measure (CORE-OM) is a self-report questionnaire formed by 34 items (18 in the reduced version) that aims to assess the clinical state of a person in the following four dimensions: subjective well-being, problem/symptoms, general functioning and risk [40]. Trujillo et al. [41] studied the psychometric properties in the Spanish adapted version of the test and found an acceptable to excellent internal consistency range for all clinical dimensions (α =  0.73–0.94) and a satisfactory test–retest reliability for all domains (ρ  =  0.76–0.87), except for the Risk scale (ρ  =  0.45).

#### 3.2.7. Category System to Code Personal Constructs (CSPC)

The Category System to Code Personal Constructs (CSPC) coding system has been designed within our research group and has been used in different studies in English (e.g., [36]) and Spanish to code the personal constructs elicited in the RGT in six content areas broken down into 45 categories. Feixas et al. [42] found a high reliability index for the CSPC (K = 0.93).

### 3.3. Procedure

At its onset, this research project received approval from the bioethical committee boards of the Universitat de Barcelona and the participating centers. Professionals working at these mental health centers were asked to inform patients with a clinic profile or verified diagnosis of BPD about the existence of the present study and to invite them to participate.

In general, we expect to complete a series of two assessment sessions (or three sessions if the patient needs more time) in order to explain the nature of the study and administer all necessary instruments. We seek to verify the viability of a BPD diagnosis first (SCID-II and DIB-R), and then to obtain clinical measures (DASS-21 and CORE-OM). Finally, if the diagnosis is compatible with the requirements of the study and no exclusion criteria are met, participants proceed to complete the RGT.

A year after the first contact, we ask participants to complete a follow-up assessment in order to collect data in regard to current clinical state (DASS-21 and CORE-OM) as well as relevant treatment and mental health information, such as psychiatric emergencies and life-threatening crises during the last year.

### 3.4. Data Analysis

All data collected with the RGT will be analyzed with the program GRIDCOR 6.0 (Universitat de Barcelona, Barcelona, Spain), [43] to obtain relevant cognitive indexes, including cognitive conflict indicators. All measures derived from the RGT along with the rest of the quantitative information obtained by the rest of the instruments will be introduced and statistically analyzed with SPSS v.26 (IBM, Armnok, USA).

In order to prove the first two hypotheses, chi-square tests and variance analyses will be performed to determine whether there exist differences between groups regarding the presence, number and intensity of cognitive conflicts and, secondarily, other measures derived from the RGT. According to the obtained results, a logistic regression will be conducted to determine the capacity of the studied variables to predict group allocation. In the previous tests, and depending on the variations found between samples, relevant sociodemographic and clinical variables will be statistically controlled.

Considering only the clinical sample, correlations and analysis of variance will be performed to study the relations between cognitive variables (i.e., number and intensity of cognitive conflicts) and the clinical variables such as those retrieved from CORE-OM, DASS-21 and DIB-R scales and SCID-II BPD diagnostic criteria type and amount. Linear regression may be conducted depending on these results to assess the capacity of the studied cognitive conflict variables to predict progress one year later. In addition, based on the variability found in the sample, other variables will be statistically controlled. The study of predictive capacity of the variables with respect to treatment outcome one year later will be approached using regressions.

For the content analysis, two investigators will encode the personal constructs obtained with the RGT following the CSPC in the BPD sample. Differences in the content between congruent and discrepant constructs will be explored using chi-square tests.

### 3.5. Ethical Aspects and Trial Registration

The University of Barcelona’s Bioethics Commission (IRB 00003099) and the different ethics departments and clinical directors of all participating centers have approved the present study. All researchers involved in this project will be subject to the directives established in the data protection regulation (EU) 2016/679 of the European Parliament and of the Council of 27 April 2016. This study has been successfully registered on ClinicalTrials.gov and can be found using the identifier number NCT04498104.

## 4. Discussion

Out of all personality disorders, BPD is one of the most life-threatening and psychologically distressing conditions due the symptomatic manifestation of impulsive behavior, along with emotional, interpersonal and self-image instability. Suicidality and NSSI rates in this population are extremely high and worrying for families and mental health professionals. Over recent decades, BPD has obtained broad attention and several psychological treatments have been proposed to deal with its main symptoms. Although some treatments have proven to be effective in some domains, it is clear that not all the symptoms experienced by these patients are treated successfully, and some cases seem not to improve much even when treated with the most effective therapies. 

Cognitive conflict has been defined and conceptualized by different psychological orientations, but only a few of them have included it as an influential factor in BPD and its treatment. Those that have incorporated it into the treatment have mainly focused on theoretically based conflicts, those the academics would regard as relevant or common, instead of exploring the idiosyncratic conflicts of each patient.

Being a disorder that is noticeably characterized by identity disturbances, we expect to find relevant unresolved cognitive conflicts regarding self-construction among patients with BPD. Addressing these issues could be potentially useful to further develop an explanatory model for this disorder as well as to increase therapeutic resources to help patients identify, deal with and, eventually, resolve their inner conflicts. 

The study also presents some limitations. Its design, mainly cross-sectional in nature, precludes attributing causation to any of the potential associations found among variables. Additionally, the fact that patients are included in the study in different phases of their treatment and in different therapeutic programs might be a potential confounding factor. Although all meeting good quality levels for their usual treatment, different centers (or even different therapists) might influence in various ways the outcome measurement of the study. With respect to the value of the variables studied to predict outcome, the fixed one-year period (marked by the funding framework of the study) and also the criteria used for evaluating the outcome constitute other limitations of the study. Further studies will benefit substantively from measuring the potential effects of cognitive conflicts and other baseline variables on the outcome of different well-controlled treatment conditions and at different time intervals (e.g., end of treatment, 2- and 5-year follow-up assessments).

Another limitation of the study regarding the control group is that a thorough clinical exploration (including personal or family history of BPD) was not conducted because they were volunteers with little time available. However, some clinical questionnaires were certainly administered (CORE-OM, Symptom Checklist-90-R [44] and Beck Depression Inventory-II [45]) and their scores ranged within the normal range. In sum, unfortunately a personality disorder or other disorder cannot be completely discarded in the control group of this study, although our data suggest that the possibility for this is low.

We expect that the initial findings to be obtained with this study will provide the bases for organizing (and asking for funding for) a larger longitudinal study beginning with participants at the initial stages of the development of BPD, followed by careful assessments spread out across a long period of time, long after finalization of one treatment (or several, in a controlled comparative design).

## 5. Conclusions

The aim of this study is to investigate whether BPD-diagnosed patients have a higher presence and number of cognitive conflicts (operationalized as implicative dilemmas and dilemmatic constructs) as compared to controls. In addition, we will test if these factors are associated with symptom severity and whether having these conflicts might influence psychological distress and BPD symptomatology after treatment. If the results support these ideas, it would be interesting to further investigate if this issue could be addressed with a psychotherapeutic intervention, such as dilemma-focused intervention, based on PCT, which aims to explore and deal with idiosyncratic self-construction conflicts. Due to the already partially favorable results obtained by psychological treatments such as DBT or MBT, an interesting future research line could be focused on investigating whether a dilemma-focused intervention module seeking conflict resolution would increase the positive, but still insufficient, outcomes that are already being obtained by these therapies (or even others).

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
