# Peer review of "Cognitive Conflict in Borderline Personality Disorder: A Study Protocol"

_behavsci, 2020, doi:10.3390/bs10120180_

Round 1

Reviewer 1 Report

The authors present the protocol of a study aiming to evaluate cognitive conflicts in patients with borderline personality disorder and the association between cognitive conflicts, symptom severity and treatment outcome. The topic is of relevance considering the prevalence and the severe impact of borderline personality disorder. The introduction is clear and well referenced. The study is well described, strengths and limitations are clearly stated.

I only have minor suggestions and comments:

  • At page 3, lines 136-138, the authors might report the software used for power analysis
  • At page 4, line 146: were controls screened for absence of personal / familial history of borderline personality disorders / other personality disorders or was this self-reported? The authors might also add the reference (if available) of the previous study for which controls were recruited as well as time of collection
  • How many patients, considering potential loss at follow-up, do the authors expect to be available for the new assessment after one year?
  • At page 6, lines 278-282, can the authors better detail how treatment outcome will be assessed and coded (which specific instruments will be used to define response, will there be a threshold to define responders or will response be coded as a quantitative variable?). In general, I think that a reader might find clearer if what the authors define "treatment outcome" (which treatments? response measured with which instruments?) would be more explicit also in other parts of the text (e.g. abstract, discussion) 
  • Will there be stratified analyses according to response to different treatments?

Author Response

Thank you.

Reviewer 2 Report

This is an interesting study proposal on BPD. I have the following specific comments for the authors to consider.

Specific comments:

  1. The definition of "cognitive conflict" could be more clearly laid out in the introduction section.
  2. Rather than say "It is not rare for BPD ...", it would be better to simply state that "It is common for persons with BPD".
  3. Please change "alpha risk of" to "alpha error of".
  4. It is contradictory that on one hand, subjects were diagnosed to have BPD according to DSM-5 but under the methods section, authors write that "a structured assessment interview for DSM-IV-II (SCID-II)" would be utilised.
  5. An alternative model described in DSM-5 for personality disorders includes essential features for personality disorders, with specific features added to denote the specific personality disorder, or a dimensional model. This should be at least briefly discussed.

Author Response

Thank you. 
